# Nuc2Vec: Learning Representations of Nuclei in Histopathology Images with Contrastive Loss

**Chao Feng**[1,2]                                                    FENGC@MSKCC.ORG
[1] *Department of Pathology, Memorial Sloan Kettering Cancer Center*
[2] *Program in Physiology, Biophysics and Systems Biology. Weill Cornell Medical College*

**Chad Vanderbilt**[1]                                               VANDERBC@MSKCC.ORG
**Thomas J. Fuchs**[3]                                        THOMAS.FUCHS.AI@MSSM.EDU
[3] *Hasso Plattner Institute for Digital Health, Icahn School of Medicine at Mount Sinai*

**Editors:** Under Review for MIDL 2021

## Abstract

The tumor microenvironment is an area of intense interest in cancer research and may be a clinically actionable aspect of cancer care. One way to study the tumor microenvironment is to characterize the spatial interactions between various types of nuclei in cancer tissue from H&E whole slide images, which requires nucleus segmentation and classification. Current methods of nucleus classification rely on extensive labeling from pathologists and are limited by the number of categories a nucleus can be classified into. In this work, leveraging existing nucleus segmentation and contrastive representation learning methods, we developed a model that learns vector embeddings of nuclei based on their morphology in histopathology images. We show that the embeddings learned by this model capture distinctive morphological features of nuclei and can be used to group them into meaningful subtypes. These embeddings can provide a much richer characterization of the statistics of the spatial distribution of nuclei in cancer tissue and open new possibilities in the quantitative study of the tumor microenvironment.

**Keywords:** histopathology, tumor microenvironment, nuclei subtyping, constrastive learning, representation learning, unsupervised learning

## 1. Introduction

The tumor microenvironment (TME) plays a vital role in the growth and metastasis of cancer (Arneth (2020)). One important characteristic of TME is the spatial interactions between various types of nuclei in cancer tissue, e.g., the infiltration of tumor cells by lymphocytes (Paijens et al. (2020)). Hematoxylin and eosin (H&E) stained histopathology slides provide a holistic morphological picture of TME and serve as a critical tool for the clinical assessment of cancer. Recent advances in digital and computational pathology, in particular, nuclear detection or segmentation, and nuclear classification (e.g., Xie et al. (2019); Graham et al. (2019); Sirinukunwattana et al. (2016); AbdulJabbar et al. (2020)), have enabled large scale delineation of nuclear maps in whole-slide images (WSI). These developments not only promise more efficient and reproducible clinical evaluation of TME but also opened the possibility of discovering new statistics to quantitatively characterize TME for a deeper understanding of the diseases and their prognosis.

Existing nucleus classification methods only classify nuclei into one of several categories, such as lymphocytes, tumor cells, healthy epithelial cells, etc (Graham et al. (2019); AbdulJabbar et al. (2020); Gamper et al. (2019); Diao et al. (2020)). Such a paradigm may

ignore subtle morphological differences such as those distinguishing tumors from different cancer types. A finer-grained classification could help understand the diversity of disease processes and identify clinically meaningful subgroups of patients. However, such endeavors based on current supervised nucleus classification methods would require tedious efforts of pathologists to label each nucleus as one of many categories, which may also be prone to inter-observer variations. In this work, we propose Nuc2Vec, an unsupervised method that learns vector embeddings of nuclei based on their morphological features presented in H&E images. Such embeddings can be used for fine-grained clustering of nuclei population and provide a more comprehensive description of the nuclei landscape in WSIs of cancer tissue.

Our approach is based on the contrastive learning methods that have achieved reasonable success in unsupervised classification of natural images (Chen et al. (2020b,a); Van Gansbeke et al. (2020)). Our hypothesis is that certain transformations of the image patch centered around a nucleus, such as color jitter and rotation, should not change its identity. We chose a specific version of contrastive learning loss as proposed in Wang and Isola (2020): the euclidean distance between embedding vectors from image augmentations of the same nuclear instance is minimized; otherwise, the embedding vectors are distributed uniformly on a unit high-dimensional sphere. We show that our method learned embeddings such that similar nuclei are close to each other in the embedding space (section 3.1) and these embeddings can be used to group nuclei into visually meaningful subtypes (section 3.2).

The main contributions of this work are as follows: (i) we designed Nuc2Vec based on contrastive learning and introduced 'background replacement' image transformation to enable efficient representation learning for nuclear morphology; (ii) we expanded nucleus classification by increasing the number of categories from a few to over a hundred, and conducted in-depth review of these find-grained nuclear subtypes; (iii) we conducted training and evaluation at scale by investigating one million nuclei from ten different cancer types.

## 2. Methods

### 2.1. Overview of our work

In Figure 1, we provide an overview of our methods. We first collect a dataset of nuclear instances by selecting random tiles from WSIs and perform nucleus segmentation using Hovernet (2.2). We then use Nuc2Vec to compute embeddings of nucleus images (2.3). We perform hierarchical clustering on the nuclear embeddings and extract stable flat clusters of nuclei (2.4). Finally, for each cluster we randomly select 100 nuclear instances for expert review by a board-certified pathologist to determine whether the clustering captures known or novel nuclear subtypes with distinctive morphological features.

### 2.2. Datasets of Nuclear Instances

We created a dataset of nuclear instances from WSIs of the patients who have undergone broad genomic sequencing analysis by MSK-IMPACT at Memorial Sloan-Kettering Cancer Center (MSKcc). The slides were chosen from the sequencing cohort as they are pre-selected to contain predominantly tumor tissue. We selected ten of the most prevalent type of cancers, including Bladder, Breast, Colorectal, Endometrial, Ovarian, Pancreatic,

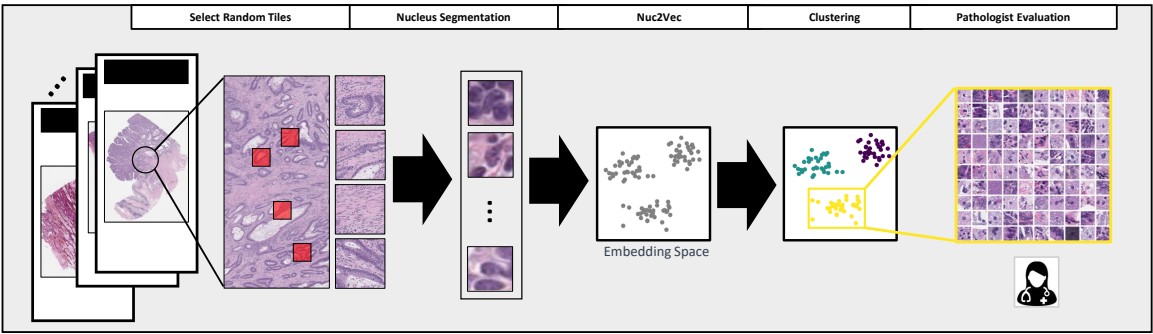

*Figure 1: Overview of Methods: We collect a dataset of nuclei through nucleus segmentation on randomly selected tiles from WSIs. We developed Nuc2Vec to compute vector embeddings for these nuclei. Hierarchical clustering of the embeddings reveals subtypes in the nuclei population which are verified by a board-certified pathologist.*

Prostate Cancers as well as Non-small Cell Lung Cancer, Glioma and Melanoma. For each type of cancer, we randomly selected 50 slides (one slide per case) among 100 slides with the largest tissue areas. All slides were scanned at 20×. For each slide, we randomly select 400 512×512 image tiles. We then run the Hovernet inference (Graham et al. (2019)) using weights provided by the author (pretrained on the Pannuke dataset (Gamper et al. (2019))) with Tensorflow (Abadi et al. (2015)). Since the weights were trained on 40× images, we resize our selected images tiles to 1024×1024. Finally, for each type of cancer, we randomly select 100k nuclear instances from all the segmented nuclei. As such we obtain a dataset of 1 million nuclear instances of ten different cancer types. For each nucleus, we define the 128×128 image patch centered around it as a nucleus image.

### 2.3. Nuc2Vec

To learn embeddings that represent the morphological essence of nucleus through contrastive learning, we need to design image transformations that perturb the nucleus image as much as possible while preserving its semantics. We use several standard image transformations including resizing with a random scale between 0.5 and 1, randomly applied color jittering, Gaussian blurring, rotation, vertical and horizontal flipping, each with 50% probability. In addition to these, to emphasize the morphological features of the nucleus itself rather than the texture of its background. We also propose a new image transformation called 'background replacement'. We generated bounding box for each nucleus from the segmentation masks predicted by Hovernet. In 'background replacement', we maintain the region of nucleus image within the bounding box while the background region is replaced by a randomly sampled 128×128 region in the same 1024×1024 image tile. As such we place the same nucleus in a different but similar context. We show in section 3.1 that without this transformation, the background of the nucleus is overemphasized, and the features of nucleus itself are not adequately accounted for by the model. We call this method of contrasive learning with 'background replacement' transformation Nuc2Vec.

We use a specific variant of MOCOv2 (Chen et al. (2020b)) and adapted code from Wang and Isola (2020) which is implemented in Pytorch (Paszke et al. (2019)). Specifically,

we use a ResNet34 as the encoder to compute a 128-dimensional embedding vector, and added another linear layer of size 512 with ReLU activation before the final embedding. We use the combined alignment and uniformity loss as described in Wang and Isola (2020), which we briefly explained in the third paragraph of section 1. All models used in the result sections are trained on the dataset described in 2.2 for 30 epochs with 4 Tesla V100 GPUs.

### 2.4. Hierarchical Clustering of Nucleus Embeddings

To discover subtypes in the nuclei population, we perform hierarchical clustering of the nuclear embeddings with ward linkage using *fastcluster* package (Müllner (2013)). We use the 'excess of mass (EOM)' algorithm (Campello et al. (2013b)) as implemented in HDBSCAN package (McInnes et al. (2017); Campello et al. (2013a)) to extract flat clusters with different levels of granularity. To evaluate the stability of clustering with respect to the 'minimum cluster size (MCS)' parameter used in EOM, we randomly select 10 subsamples of 100k nuclei from the training dataset. For each subsample, we perform hierarchical clustering, extract flat clusters using different values of MCS, and assign each instance in the training datasat by a majority vote of the cluster labels of its 1023 nearest neighbors in the subsample. For each MCS, we calculate the average pairwise adjusted mutual information score (AMS) between clustering results based on different subsamples. We found that the AMS is approximately 0.6 within a range of MCS values (300~700).

### 2.5. Assigning Clusters to New Nuclear Instances

Given a nucleus image unseen during the training of Nuc2Vec, we can compute its vector embedding using the trained ResNet34 model. We then find its 1023 nearest neighbors in the embedding space among the training data using *faiss* (Johnson et al. (2017)). Finally, we take the majority vote of the cluster labels of these 1023 nearest neighbors as the assigned cluster label for the given nucleus image.

## 3. Experimental Results

### 3.1. Nuc2Vec is Essential for Learning Useful Embeddings of Nuclei

A useful embedding should place nuclei with similar features close to each other. Thus, we randomly selected nuclear instances from the training dataset and visualize their nearest neighbours in the embedding space. To demonstrate the effect of our method, we also constructed two other embeddings for each nucleus: a 27-dimensional hand-engineered features and features learned using contrastive loss without 'background replacement'. Specifically, the hand-engineered features are constructed based on the segmentation masks predicted by HoverNet, including area, eccentricity, solidity, extent, major/minor axis length, perimeter, orientation and 9-dimensional central moments, as well as texture features such as the statistics of gray level co-occurrence matrices. These features are similiar to those used in Zhou et al. (2019). Each dimension of the feature is converted to standard score. Based on expert review, we found that Nuc2Vec is best at capturing the essence of nucleus morphology.

In Figure 2, we illustrate the quality of the embeddings with four examples chosen from 42 randomly selected samples. The four instances are marked by green boxes. For each

example, we show its 15 nearest neighbours according to three different embeddings: hand-engineered features (A), embeddings learned using contrastive loss without 'background replacement' (B) and Nuc2Vec (C). From left to right, the first instance is a spindle-shaped stromal nucleus; the next two instances are tumor nuclei with visible nucleolus, with the second one slightly bigger; finally we have a tumor nucleus with white content in the center, which suggests the complexity of nucleus as indicated by a lack of Hematoxylin uptake. In all four instances, the nearest neighbours according to embeddings learned by Nuc2Vec most clearly captures said nucleus morphology.

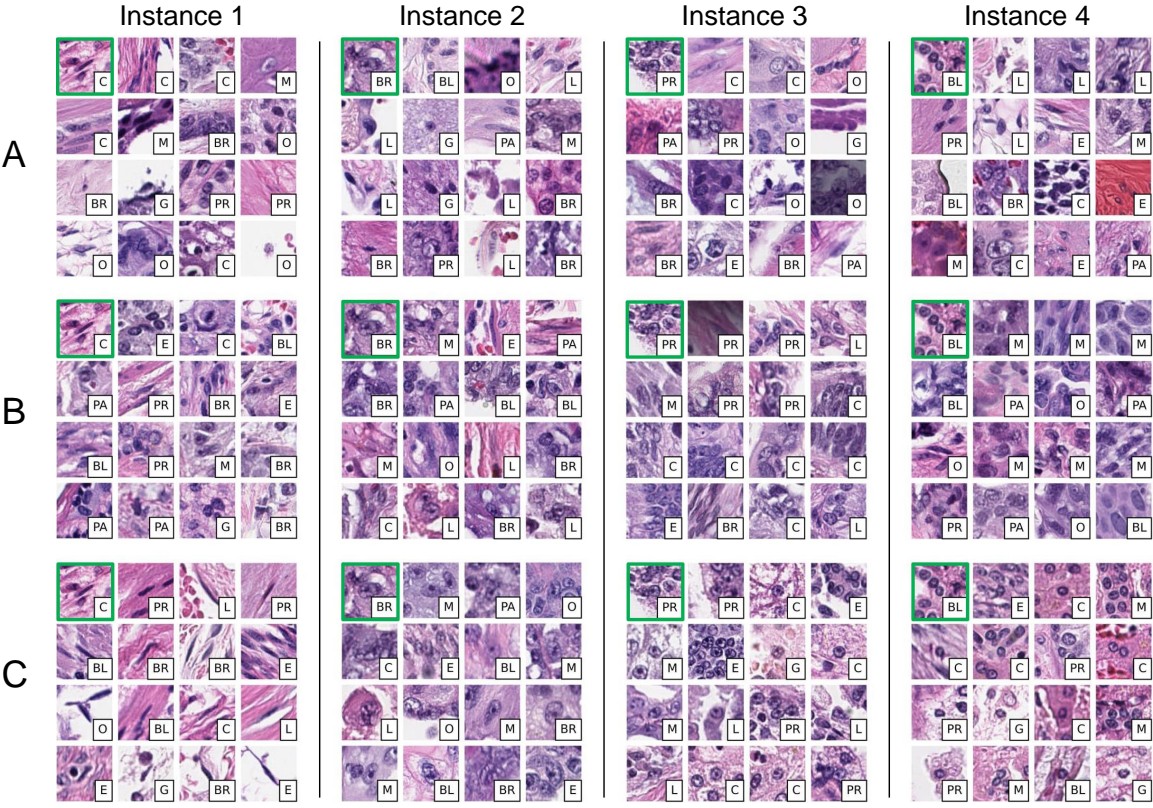

*Figure 2: Comparison of embeddings based on the 15 Nearest Neighbors of four randomly selected nuclei instances: hand-engineered features (A), contrastive learning without 'background replacement' (B) and Nuc2Vec (C). Letters in the bottom right box of each nucleus image refer to the tissue type as follows, BL: Bladder; BR: Breast; C: Colorectal; E: Endometrial; G: Glioma; L: Lung; M: Melanoma; O: Ovarian; PA: Pancreas; PR: Prostate.*

## 3.2. Hierarchical Clustering Uncovers Subtypes in Nuclei Population

We use hierarchical clustering of the learned embeddings to identify subtypes in the nuclei population. For each subtype, we randomly sample 100 nucleus images for a board-certified pathologist to review their morphological features and determine if a majority of them belong to a known nucleus type. In Figure 3, we show a dendrogram of clustering result based on a randomly selected subsample of dataset as described in section 2.4. We use

EOM with MCS equals 400 (for a dataset of size 100k), which extracted 140 clusters. Note that different MCS does not affect the stability of clustering results significantly. On the other hand, a larger value of MCS will lead to a smaller number of clusters and vice versa. We choose this specific value for qualitatively optimal clustering.

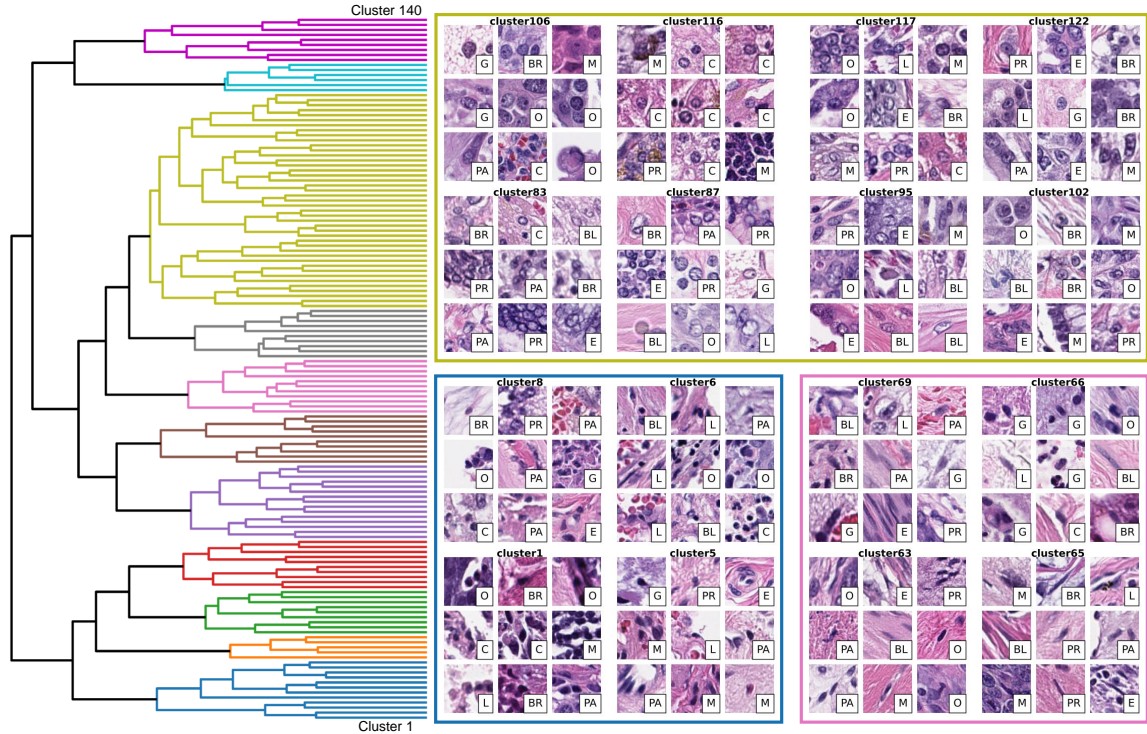

Figure 3: Dendrogram of the hierarchical clustering result using a randomly selected subsample of 100k nuclear instances. Each leaf corresponds to one of the 140 clusters extracted with the EOM algorithm. The clusters are ordered from 1 to 140 from bottom to top. We organize these clusters into 11 branches. Nuclei samples from selected clusters are depicted for the blue, pink and olive branch in the boxes with the corresponding color. See text for a detailed description of the morphology features of these branches. Tissue types are coded as in the previous figures.

We can organize the 140 clusters into 11 branches by cutting the hierarchical clustering tree at a fixed threshold. Through expert review, we found that most of these branches have well defined features while nuclei are further grouped into clusters within each branch with more subtle morphological differences. For example, as shown in Figure 3, the blue branch (clusters 1-12) are predominately lymphocytes; the pink branch (clusters 62-72) are mostly stromal or elongated tumor nuclei. Most interestingly, the largest, olive branch (clusters 83-125) are mostly composed of diverse clusters of tumor nuclei with distinctive features. To partially verify the morphological features of each cluster, a board-certified pathologist provided scores in terms of size, darkness, color consistency, border irregularity, cytoplasm visibility as well as degree of elongation of the nuclei, based on visual inspections of 100 randomly selected samples for each cluster. We provide more detailed discussion

of the morphological scores in the appendix and include the score table as supplemental materials.

For each cluster, we calculated the entropy of the distribution of their occurrence in the ten different cancers. In Figure 4, we show randomly selected nucleus images from each of the 6 clusters with the lowest entropy. Clusters 49 and 50 are predominately nuclei from Glioma, which reflects the fact that Glioma is a distinct non-epithelial malignancy with unique cytomorphology. Almost half of the nuclei in cluster 52 are from Endometrial cancer. These nuclei are mostly from stromal cells in muscle tissue, which accounts for a large portion of Endometrial tissue. Colorectal cancer is the major type for cluster 116. These nuclei appear to be surrounded by mucins, which is indicative of cancers from the GI tract. Finally, cluster 14 and 15 seems to be predominately cancers from female organs (breast, ovaries and uterus).

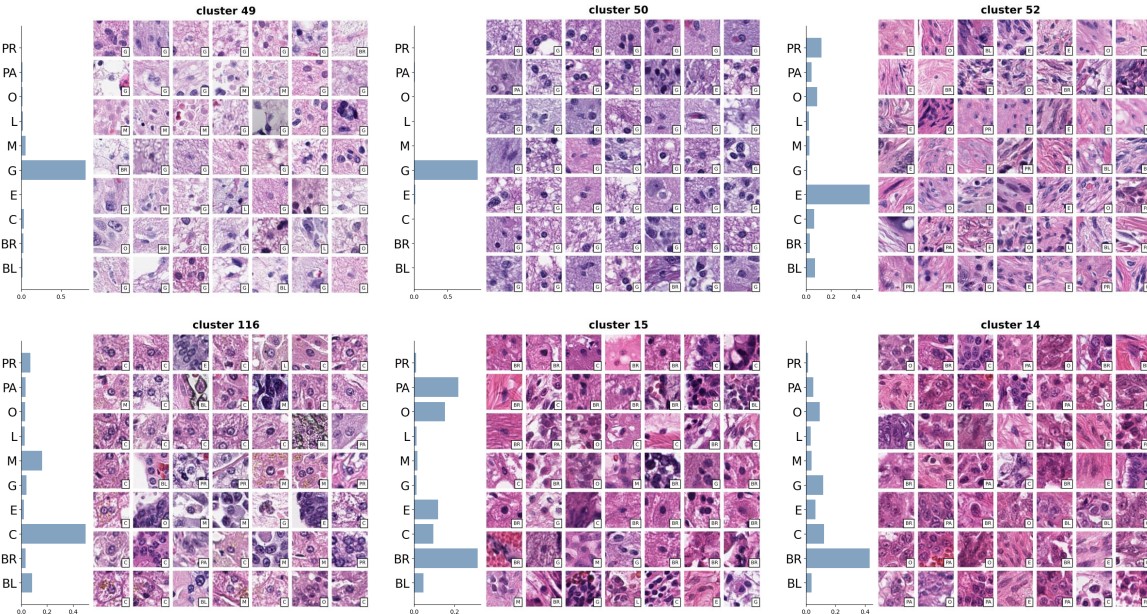

Figure 4: Example nuclei from the six clusters with the most imbalanced distribution of the ten cancer types. The distribution is shown as bar plot on the left of each image. BL: Bladder; BR: Breast; C: Colorectal; E: Endometrial; G: Glioma; L: Lung; M: Melanoma; O: Ovarian; PA: Pancreas; PR: Prostate. See text for detailed description for each cluster.

### 3.3. Assigning Cluster Labels to a New Dataset

To test the generability of the clusters discovered by Nuc2Vec to nuclear instances unseen during training process, we collected a much larger dataset of over a billion nuclear instances from 991 slides of the ten cancer types in the same cohort. We assign the cluster label to each instance of the dataset using the procedure described in section 2.5. In Figure 5 we compare sample nuclei from four clusters for both the dataset used for training and the new dataset. Qualitatively, our procedure is able to assign a give nuclear instance to the cluster that best captures its morphological essence.

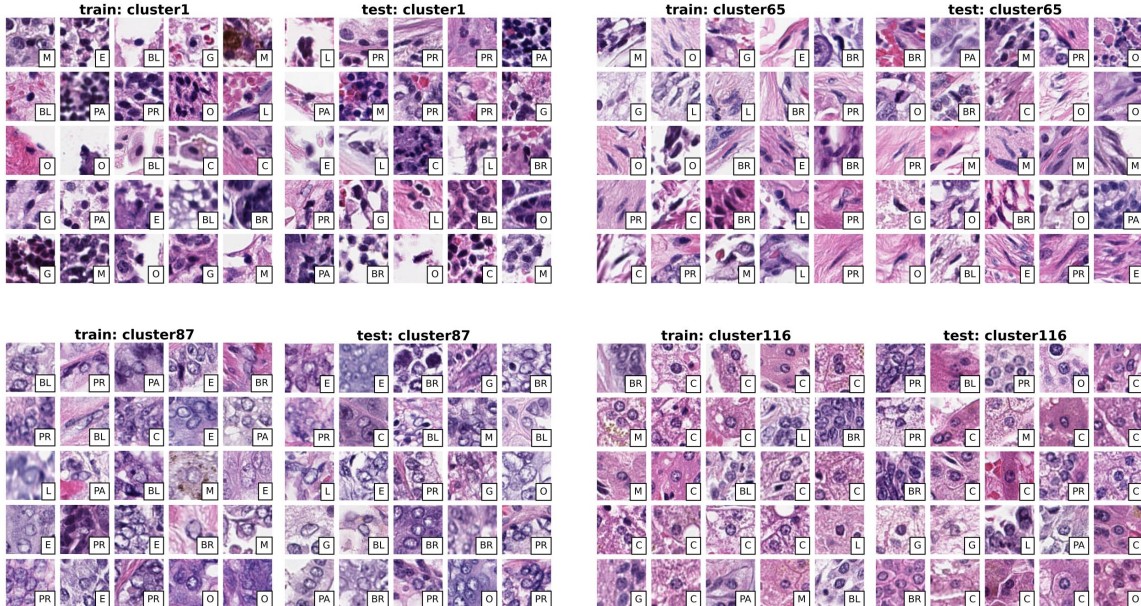

*Figure 5: Sample nuclei from selected four clusters for both dataset used for training (train) and the new dataset (test). Tissue types are coded as in previous figures.*

## 4. Conclusion and Future Works

The the best of our knowledge, we have presented the first system of at scale representation learning for nuclei in H&E stained histopathology images. Although our method rely on previous supervised nucleus segmentation model, it does not require extensive annotations of nucleus subtypes. The embeddings learned by Nuc2Vec are able to capture fine-grained morphological distinctions among subtypes in nuclei population and has the potential for providing deeper understanding of the disease process and more accurate prognosis analysis.

Although most of the evaluations we presented are qualitative, we emphasize that histopathology is to a large extent an empirical discipline. We aim to further validate our results in future works through two veins: 1) examine our nucleus subtyping against orthogonal technologies that applied to cancer tissue, such as spatial transcriptomics (Levy-Jurgenson et al. (2020)); 2) assess the value of the fine-grained subtyping by using them to construct spatial statistics features of tumor microenviroment in WSIs and perform downstream clinical tasks such as molecular biomarker prediction (Diao et al. (2020)).

### Conflict of Interest

T.J.F. is a co-founder and equity holder of Paige.AI. C.F. and T.J.F. have intellectual property interests relevant to the work that is the subject of this paper. MSKcc has financial interests in Paige.AI and intellectual property interests relevant to the work that is the subject of this paper.

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

## Appendix A. Partial Verification of Morphological Features for Each Nucleus Clusters

To partially verify the morphological features for each of the 140 clusters, a board-certified pathologist provided score in terms of size, darkness, color consistency, border irregularity, cytoplasm visibility as well as degree of elongation of the nuclei, based on visual inspections of 100 randomly selected samples for each cluster. These are fundamental nucleus morphological features used by pathologist to classify nuclei into high level categories such as lymphocytes and tumors. To partially blind the pathologists from the similarities between nearby clusters in the hierarchical clustering tree, we randomly shuffled the ordering of clusters for the sample nucleus images. The scoring table is included as the supplemental materials. These scores qualitatively verifies that most clusters discovered by Nuc2Vec captures distinct morphology in terms of these fundamental features. We want to emphasize, however, that these features alone does not constitute a good representation of the nuclei. Indeed, the hand-engineered features we designed in section 3.1 captures almost all aspects of these fundamental features (other than cytoplasm visibility). However, our qualitative comparison in Figure 2 shows that the Euclidean distances of these hand-engineered features are unable to represent the similarities between nuclear instances. Contrastive loss provides a natural way of learning representations of nuclei, and hence the proper similarity metrics between them, which can then be used for fine-grained clustering.

