# OpenReview forum: "Nuc2Vec: Learning Representations of Nuclei in Histopathology Images with Contrastive Loss"
_MIDL.io/2021/Conference — MIDL 2021_

### Official Review · AnonReviewer3 · 2021-02-26

**Confidence:** 5
**Preliminary Rating:** 4
**Recommendation:** Oral
**Final Rating:** 4

**Summary:**

The paper described a novel method for clustering segmented nuclei in histopathological images. Although the segmentation part was done in a supervised manner (using a pre-trained HoverNet model), embedding and clustering were performed with unsupervised approaches. A big dataset containing 1 million nuclei were used in this work and the evaluation was done qualitatively on a subset of the dataset. The results showed that the proposed approach delivered acceptable clustering results for known nuclei classes and could be potentially used for clustering nuclei to unknown and new classes.

**Strengths:**

- The proposed approach is novel and can be very helpful for the research community in the field.
- The paper is well-written and easy to follow in all parts.
- State-of-the-art methods were used in different parts of the proposed method (for segmentation, embedding and clustering)

**Weaknesses:**

- The results were only qualitatively analysed. It would be interesting to investigate the performance of the presented method on the publicly available datasets that provide nuclei segmentation and classification masks. Examples of such datasets set can be found below:
MoNuSAC dataset: https://monusac-2020.grand-challenge.org/
CoNSeP dataset: https://warwick.ac.uk/fac/cross_fac/tia/data/hovernet/


**Deanonymize Review:**

no

**Detailed Comments:**

- It is not clear why bounding boxes with 128X128 pixels were chosen to sample the nuclei. Would it possible to decrease the dimension (e.g. 64x64 pixels) to concentrate more on the nuclei and remove the background noise?

- While the proposed "background replacement" approach is interesting and the authors showed its effectiveness in the results section, wouldn't be better just to replace the background with black pixels? The potential problem with the proposed "background replacement" method is that it is possible that other nuclei, from the randomly selected part of the image, can appear in the replaced background and thus affect the clustering and embedding step negatively.

- For better clarification, it is suggested to mention that the segmentation part is done in a supervised method using the pre-trained HoverNet model.

**Final Rating Justification:**

I thank the authors for addressing all my comments and I hope in the future they can conduct those additional experiments to see how they affect the results. I still think the paper is absolutely worth publishing in the MIDL 2021.

**Justification Of The Preliminary Rating:**

The paper presents a novel approach for nuclei clustering in an unsupervised manner that could be very helpful for the research community in the field.  Moreover, the paper is well written and the results section include very interesting observations. Although the paper quality can be still a bit improved (refer to the "detailed comments" and "weaknesses" section), I strongly recommend this work for publication in MIDL 2021.

**Paper Type:**

both

**Special Issue:**

yes

---

> ### Author Response · Authors · 2021-03-18
> **Response to AnonReviewer3**
>
> We thank AnonReviewer3 for the kind review and excellent suggestions. Below are our detailed responses.
>
> AnonReviewer3 suggested in the Weaknesses section that we could evaluate the performance of Nuc2Vec for classification tasks on a publicly available dataset such as MoNuSAC and CoNSeP. Although we want to emphasize the main goal of Nuc2Vec is to discover new (potentially unknown or not well characterized) nuclei subtypes, we do agree that in principle the learned embedding should be able to classify nuclei into known classes. We propose two ways to evaluate this hypothesis: 1) we can use the model trained by Nuc2Vec as a pre-trained model and fine-tune a supervised classifier; 2) we can conduct an unsupervised clustering on the embeddings learned by Nuc2Vec (i.e. extract flat clusters from hierarchical clustering, or assign each nucleus to one of the clusters discovered using our dataset with the procedure described in Section 2.5 of the updated manuscript). The latter approach is more desirable as it can potentially be applied to datasets with limited labels. In [1], it is argued that to accurately classify nuclei, we may need a larger context than just the local image patch. Indeed, from one of us (a pathologist)'s experience, it is sometimes difficult to distinguish between, say a healthy epithelial cell and lymphocytes from their local context alone. We believe applying Nuc2Vec to publically available datasets with ground truth nuclei classification labels can potentially help identify different scenarios where such difficulty arises.  Unfortunately, we may not be able to conduct detailed experiments and discussions for the current version of the paper. We plan to conduct these experiments for a future version of the paper.
>
> The comments provided by AnonReviewer3 in the first two points of the Detailed Comments sections are very interesting. However, 128X128 is actually the minimum size we could use since the biggest nuclei in our training dataset has a bounding box (determined by the segmentation mask predicted by Hovernet) of size 127. We are currently conducting additional experiments to evaluate the effect of 'background removal' on the quality of learned embeddings as suggested by AnonReviewer3.
>
> We will modify the paper to emphasize our approach is only unsupervised in terms of learning the vector embedding for nuclei but relies on a supervised nucleus segmentation algorithm (HoverNet).
>
> [1] Gamper, Jevgenij, et al. "Pannuke dataset extension, insights and baselines." arXiv preprint arXiv:2003.10778 (2020).

---

### Official Review · AnonReviewer4 · 2021-03-06

**Confidence:** 5
**Preliminary Rating:** 3
**Recommendation:** Poster

**Summary:**

In the paper "Nuc2Vec: Learning Representations of Nuclei in Histopathology Images with Contrastive Loss" authors proposed a method to describe nuclei by vectors, that can be used to group them into subtypes. The presented experiments are well designed. The achieved results are interesting. The proposed method can be used to perform new studies in the biomedical domain.

**Strengths:**

The proposed method is well described and easy to follow. In the experiments, the authors used various types of tumor, which present that method can be applied for many organs/cancer types. This method can be used for future research on cell biology/ cancer biology.

**Weaknesses:**

The main weakness is a limited novelty, but the paper is interesting and can be useful for the research community.
Other comments:
1. the images presented on figures are significantly too small, nothing is visible,
2. captions used on figures should be readable, now they are too small.

**Deanonymize Review:**

no

**Justification Of The Preliminary Rating:**

The main weakness is a limited novelty, but the paper is interesting and can be useful for the research community. The proposed method is well described and easy to follow. In the experiments, the authors used various types of tumor, which present that method can be applied for many organs/cancer types.

**Paper Type:**

both

**Questions To Address In The Rebuttal:**

Is this method publicly available?

**Special Issue:**

yes

---

> ### Author Response · Authors · 2021-03-18
> **Response to AnonReviewer4**
>
> We thank AnonReviewer 4 for the kind review.
>
> With regard to novelty, we would like to emphasize that all existing nucleus classification methods in computational pathology are supervised approaches and only classify nuclei into one of several categories. To the best of our knowledge, our work is the first unsupervised approach for fine-grained clustering of nuclei based on their morphological features presented in histopathology images.
>
> We acknowledge the problems with the legibility of our figures. We will work on improving these figures.
>
> We will make our code publicly available in the coming days. Unfortunately, we cannot immediately release our training data set, but we will release the model trained on our data for the research community.

---

### Official Review · AnonReviewer2 · 2021-03-07

**Confidence:** 5
**Preliminary Rating:** 3
**Recommendation:** Poster
**Final Rating:** 3

**Summary:**

In this work, the authors leverage contrastive representation learning methods to develop a framework that learns vector embeddings of nuclei, from multiple disease sites, based on their morphological characteristics. The results presented identify 140 clusters with distinct nuclei characteristics that may be clinically useful in the diagnostic process.

**Strengths:**

The proposed method can be a very useful tool to identify, in an unsupervised fashion, similar characteristics in groups of nuclei and can serve as an important tool in potentially defining morphology-driven ground truths.

**Weaknesses:**

1. The evaluation of the presented methodology is, for the most part, qualitative in nature - Various similarity evaluation metrics have been proposed in computer vision that can be leveraged here.
2. How do the authors propose to evaluate an unseen patch and fit it to one of the N identified clusters?
3. Though the task is mostly motivated by 'lack of enough training samples, it is not clear if the identified groups are actually driven by the underlying nuclei morphology or some other characteristics. Additionally, how important are these features in the actual diagnostic process?

**Deanonymize Review:**

no

**Detailed Comments:**

1. What do the authors mean by 'textual' features?
2. Fig 2 needs quantitative evaluation. It is unclear whether the hand engineered features are specifically designed to capture the 'morphological properties' which is claimed as a basis for the clustering. In other words, if the hand engineered features were tasked to extract features such as spindle shape, capturing tumor nuclei with white center etc, will the identified neighbors be similar to that in (C)? This is a critical step for fair comparison. The arguement against doing this would be the time involved in getting substantial ground truth. However, looks like the pathologists are doing this in any case for qualitatively evaluating the results.
3. The authors use ResNet34 to compute the embedding vector. Was any modifications made to the standard architecture?
4. What are the 27 hand-engineered features computed here? It is not clear if the embedding process is similar for these features as well.



**Final Rating Justification:**

The authors have addressed most of the raised concerns.  They are currently working on including a table to assign scores in terms of different nuclei characteristics. One aspect that is still unclear is regarding the claim of difficulty in designing hand-engineered features apriori. There is a significant body of work/available resources in digital pathology regarding design and validation of such features.

**Justification Of The Preliminary Rating:**

This is an important contribution in digital pathology towards learning representations at scale. However, it is important to discuss how unseen data will be evaluated using this method.  Besides, the features considered for the hand-crafted learning are not exhaustive, and having a more expanded list is crucial to ensure a fair comparison.

**Paper Type:**

both

**Questions To Address In The Rebuttal:**

1. Would be informative if a table is provided (maybe in the supplementary material) regarding the #categories discovered. Also, how were the definition of the categories sestablished in the first place?
2. "..ignore larger morphological variations among nuclei" --> What morphological variations are being discussed here?
3. Does color normalization affect results? There is a high likelihood that will. Since this is one of the preprocessing steps in many digital pathology applications, it will be interesting to include a short discussion regarding this.
4. "These embeddings can provide a much richer characterization of the statistics of the spatial distribution of nuclei in cancer" --> It is unclear how this can be attained. Do the authors propose using these embeddings directly in learning of spatial metrics, or is this just meant to identify groups with similar properties before any downstream analysis?
5. How do the perturbation of clustering parameters affect the identified phenotypes? A discussion regarding this will be helpful to the readers.



**Special Issue:**

no

---

> ### Author Response · Authors · 2021-03-18
> **Response to AnnoReviewer2**
>
> We thank AnnoReviewer2 for the kind review and insightful comments. Below are our detailed responses for each section of Reviewer2's comment.
> ## Weakness:
> 1. Reviewer2 suggests that we could conduct a quantitative evaluation of Nuc2Vec by leveraging image similarity metrics proposed in computer vision, such as SSIM, cross-correlation, etc. However, we would like to kindly point out that Nuc2Vec is in fact learning a metric that is more suitable for representing the similarities between nuclei as presented in H&E images. This is the Euclidean distance between the vector embeddings of nuclear instances. We will update our paper to clarify this contribution of Nuc2Vec. In principle, we could compare these aforementioned metrics with the one learned through Nuc2Vec by either visually inspecting nearest neighbors of randomly selected nuclear instances, as defined by these metrics, similar to what we did in Sec. 3.1; or conduct hierarchical clustering using these metrics and visually evaluate the clustering results, similar to what we did in Sec. 3.2. However, these evaluations are still qualitative and rely on the expert knowledge and experience of a board-certified pathologist. Due to the complexity and variations of histopathology images, we do not expect these metrics to perform as well as Nuc2Vec (which is trained with the contrastive learning objective to ignore irrelevant variations) in terms of these qualitative measures. However, we are currently implementing efficient ways to compare Nuc2Vec with some of these metrics. In summary, our aim here is to develop a representation learning method to capture fine-grained differences of nuclei in H&E images. Ultimately, we wish to evaluate the usefulness of these representations in various diagnostic and prognostic tasks. But as the first step in such studies, we argue that qualitative evaluations by pathologists are essential and more informative than quantitative evaluations.
> 2. We briefly described the procedure to assign cluster labels to an unseen patch at testing time in Sec. 2.4. However, to make it more clear, we have updated our paper to include Sec. 2.5: Assigning Clusters to New Nuclear Instances. We also added Sec. 3.3 to show the results of assigning cluster labels to a new dataset.
> 3. In Sec 3.2, we have briefly explained that nuclei in the same clusters are determined by a board-certified pathologist to share similar nuclear morphological features, such as clusters identified as predominantly lymphocytes. We are currently working on a table to assign scores in terms of nucleus size, darkness, color consistency, border irregularity, elongation as well as cytoplasm visibility for all the clusters. These are the fundamental features used by a pathologist to classify nuclei into neoplastic vs non-neoplastic, as well as distinguishing various forms of inflammatory and stromal cells, etc.  We will update our paper to clarify these criteria used by one of us (a pathologist) to qualitatively evaluate and characterize the clusters of nuclei.
> In terms of importance, our long-term goal is to use these fine-grained nuclei descriptors for a wide-range of diagnostic and prognostic tasks ranging from cancer subtyping and outcome prediction such as survival analysis.  We would like to emphasize that the criteria we used to characterize these clusters are fundamental nuclear morphological features known to be relevant in diagnostic and prognostic tasks. Our contribution is a principled approach to define fine-grained subtypes of nuclei that share similarities in these features, as such to enable quantitative evaluation of their prognosis values.
>
> ## Detailed Comments:
> 1. It should be 'texture'. We have updated the pdf to provide more details on hand-engineered features in Sec. 3.1.
> 2. We would like to emphasize that one of the main contributions of Nuc2Vec, and the advantage of the contrastive representation learning method combined with hierarchical clustering based on the learned embeddings, is the ability to discover nuclei subtypes that share similar morphology features without defining these features definitively and extensively in advance. Although in Sec. 3.1 we demonstrated this ability with four examples whose morphology features have clear verbal descriptions, the algorithms can discover potentially unknown or not well-characterized nuclei subtypes, as discussed in Sec 3.2. On the other hand, it is difficult to design exhaustive hand-engineered features a priori. The list of hand-engineered features we chose, which we have included more details in Sec. 3.1 of the updated pdf, is similar to what has been used in previous research, e.g., [1].
> 3. We have updated the pdf to include more details on the ResNet34 used in Sec. 2.3.
> 4. See 1
>
> [1]Zhou, Yanning, et al. "Cgc-net: Cell graph convolutional network for grading of colorectal cancer histology images." Proceedings of the IEEE/CVF International Conference on Computer Vision Workshops. 2019.

---

> > ### Author Response · Authors · 2021-03-18
> > **Reponse to AnnoReviewer2 (Continued)**
> >
> > ## Questions to Address In the Rebuttal:
> > 1. We are currently working on a table to assign scores in terms of nucleus size, darkness, color consistency, border irregularity, elongation as well as cytoplasm visibility for all the clusters. (See our response to Weaknesses 3 above). That said we do not claim that all clusters match known morphological descriptions from pathology textbook. Our hope is a large number of newly discovered subtypes can be useful for downstream clinical tasks.
> > 2. We have modified the sentence to clarify with an example " ... [ignore] subtle morphological differences such as those distinguishing tumors from different cancer types."
> > 3. We did not directly study the effect of color normalization on Nuc2Vec. However, the image transformations used in creating two views for the contrastive objective included color jittering. The effect of color augmentation versus color normalization in training the convolutional neural network for histopathology images has been studied in previous work [2]. We argue that for the contrastive learning framework, it is more natural to use color jittering as part of data augmentation (i.e., image transformation), rather than using color normalization. We will update our paper to include a discussion of this as well as more details on the image transformations we used.
> > 4. There could be many ways to utilize the fine-grained nuclear descriptor. For example, we could calculate the likelihood of nuclei from a certain cluster occurring in the neighborhood of nuclei from another cluster from a large collection of tissue samples. Such spatial statistics could provide not only more insights into the nature of the discovered nuclei cluster but also representations for the whole slide potentially useful for downstream clinical tasks. We will update our paper to include more discussion on this in the "Conclusion and Future Works" section.
> > 5. As explained in Sec. 2.4, we use the 'excess of mass' algorithm to extract flat clusters from the hierarchical clustering tree, which has one parameter, namely the "minimum cluster size (MCS)". There we briefly discussed the effect of this parameter value on the stability of the clustering results. On the other hand, a larger value of MCS will lead to a smaller number of clusters and vice versa. We chose MCS equals 400 (for a dataset of size 100k) for qualitative optimal clustering, which we have stated in Sec. 3.2 but have now updated the pdf to include a slightly more detailed discussion.
> >
> > [2] Tellez, David, et al. "Quantifying the effects of data augmentation and stain color normalization in convolutional neural networks for computational pathology." Medical image analysis 58 (2019): 101544.

---

### Author Response · Authors · 2021-03-18
**General Responses to All Reviewers**

We thank all three reviewers for the kind reviews and insightful comments. We would like to provide two general responses to all reviewers:

1. We have uploaded a rebuttal revision of the pdf. The changes we made for this version is detailed in the individual responses to each reviewer;
2. We will make our code publicly available in the coming days. We currently are unable to release our dataset, but we will release the model trained on our dataset for the research community.

---

### Meta-Review · Area_Chair1 · 2021-03-27

**Recommendation:** Accept (Oral)

**Metareview:**

Three knowledgeable reviewers recommend accept and maintained their rating after the rebuttal and discussion. All of them agreed that the paper will be very interesting for the research community in the field.  The authors addressed the points raised by the reviewers during the discussion and updated their manuscript. Moreover, the authors said that they will share code and trained model for their submission. I think that this paper will be a good contribution to MIDL 2021. Authors should address the main points in the reviews when preparing a final version.

**Paper Type:**

both

---

### Decision · Program_Chairs · 2021-03-31

**Decision:**

Accept

**Comment:**

Congratulations your paper has been selected as a long oral.